# Transcriptomic Analysis Provides Insights into the Differential Effects of Aluminum on Peanut (*Arachis hypogaea* L.)

**DOI:** 10.3390/genes13101830

**Published:** 2022-10-10

**Authors:** Gegen Bao, Shengyu Li, Qi Zhou, Umair Ashraf, Jingxuan Qiao, Xiaolin Li, Xiaorong Wan, Yixiong Zheng

**Affiliations:** 1Guangzhou Key Laboratory for Research and Development of Crop Germplasm Resources, Innovative Institute for Modern Seed Industry, Zhongkai University of Agriculture and Engineering, Guangzhou 510225, China; 2Department of Botany, Division of Science and Technology, University of Education, Lahore 54770, Pakistan

**Keywords:** aluminum, peanut, soluble sugar, ABA, transcription factors

## Abstract

In acidic soils, high concentrations of aluminum ions (Al^3+^) in dissolved form reduce root growth and development of most crops. In addition, Al^3+^ is also a beneficial element in some plant species in low concentrations. However, the regulatory mechanism of the growth and development of peanut (*Arachis hypogaea* L.) treated with different concentrations of Al^3+^ has been rarely studied. In this study, peanut seedlings were treated with AlCl_3_.18H_2_O in Hoagland nutrient solution at four different concentrations of Al^3+,^ i.e., 0 (pH 6.85), 1.25 (pH 4.03), 2.5 (pH 3.85), and 5 (pH 3.69) mmol/L, which are regarded as Al0, Al1, Al2, and Al3. The results showed that low concentrations of Al treatment (Al1) promoted peanut growth, while high concentrations of Al treatments (Al2 and Al3) significantly inhibited peanut growth. Compared with the control (Al0), transcriptome analysis showed that the differentially expressed genes (DEGs) of starch and sucrose metabolic pathways were significantly enriched at low concentrations, i.e., Al1 treatment, whereas the expression of *AhERD6* (sugar transporter) was significantly up-regulated, and the soluble sugar content was significantly increased. The DEGs of the plant hormone signaling transduction pathway were significantly enriched at high concentrations of Al2 and Al3 treatments, whereas the expression of *AhNCED1* (9-cis-epoxycarotenoid dioxygenase) was significantly up-regulated, and the content of ABA was significantly increased. Moreover, the expression of transcription factors (TFs) in peanut was affected by different concentrations of Al. Overall, low concentrations of Al1 promoted peanut growth by increasing soluble sugar content, while high concentrations of Al2 and Al3 inhibited the growth of peanut, induced *AhNCED1* gene expression, and increased endogenous ABA content. For peanut, the exposure of Al at low concentrations not only derived an adaptive mechanism to cope with Al stress, but also acted as a stimulator to promote its growth and development.

## 1. Introduction

Aluminum (Al) is the most abundant metallic element on Earth, accounting for 7.45% of total mineral weight [1]. Usually, Al in soil mainly exists in the form of stable aluminum oxides and aluminosilicates, which are not easily absorbed by plants [2]. When soil is acidified, H^+^ will replace Al in soil minerals to form active Al that reacts with other components in the soil to form different forms of Al, e.g., exchangeable Al (EXAl), hydroxy adsorbed Al (HyAl), organic complex Al (OrAl), iron oxide bound Al (DCBAl), interlayer Al (InAl) and amorphous aluminosilicates, and gibbsite (NcAl) [3]. Among them, EXAl and HyAl are toxic to plants [4,5], whilst EXAl is the most active free Al compound in soil. When soil pH is strongly acidic, EXAl mainly exists in the form of A1^3+^, as it is the most easily adsorbed by negative charges, thereby entering plant cells [6]. When plants accumulate excess A1^3+^, the most obvious symptom is an inhibited root system [7]. Under Al stress, uptake of essential elements such as magnesium (Mg), phosphorus (P), iron (Fe), and molybdenum (Mo) through roots is inhibited, resulting in stunted plant development, reduced leaf area, yellowing, and ultimately, death of plants [8]. Moreover, P utilization efficiency and Fe transport efficiency with Al tolerance may be adapted to acidic soils, implying that the molecular mechanisms behind their co-evolution are complex [9,10].

Previously, low concentrations of Al^3+^ have been shown to have beneficial effects on a number of plant species, especially those native to the tropics in acidic soils [11]. Low concentrations of Al^3+^ alleviate H^+^ and P toxicity under low pH conditions and excess P, respectively [12]. Al^3+^-promoted growth has also been observed in low-P conditions and in plants tolerant to low pH, which means that the phenomenon that Al^3+^ promotes plant growth also exists in the absence of excess H^+^ and P [13]. Low concentrations of Al^3+^ in rice promoted growth and increased soluble sugar content in roots [11]. Soluble sugars might act as osmo-protectants, and low concentrations of Al^3+^ induced the expression of a sugar transporter (ERD6), which facilitated the transport of soluble sugars to specific tissues or organelles [11]. Thus, low concentrations of Al^3+^ act as bio-stimulants in crops to promote growth and development. However, there is no evidence whether the peanut generally grown in South China has also evolved such mechanisms to promote growth under low concentrations of Al^3+^.

Plants generally protect themselves against higher Al^3+^ concentration through external exclusion and/or internal protection mechanisms [14,15]. The external exclusion mechanism induces the production of organic acids such as malic acid, citric acid, and oxalic acid to combine with Al^3+^ to form a non-toxic complex, thereby preventing Al^3+^ from entering the root tip cells of plants [16]. The internal tolerance mechanism involves the adsorption of Al^3+^ by hemicellulose and pectin in the cell wall and the transfer of Al^3+^ into the vacuole by organic acid transporters in the cytoplasm [14]. There are two main families of transporters associated with organic acid secretion, multi-drug and toxic compound extrusion (MATE) and aluminum-activated malate transporter (ALMT). MATE and ALMT were reported to control citrate and malate secretion, respectively [9]. *AhFRDL1* encoding MATE family proteins have been cloned in peanut. Al induced the expression of the *AhFRDL1* gene and the secretion of citrate in peanut root tips, and *AhFRDL1* was involved in peanut aluminum tolerance by regulating citrate secretion [10]. Al stress-induced *ALMT* expression has been demonstrated in a variety of plants [17]. *TaALMT1*, *AtALMT1*, *GmAMLT1* in wheat (*Triticum aestivum* L.), *Arabidopsis thaliana*, and soybean (*Glycine max*) are involved in malate secretion, whereas the expression levels of these genes are higher in resistant cultivars than in susceptible cultivars [18]. *AhALMT9* in peanut has high homology with *AtALMT1* in *Arabidopsis* [15]. The cell wall prevented Al^3+^ from entering cells by inducing the expression of pectin methylesterase (PME) family genes and enhancing the activity of the PME enzyme [19]. Furthermore, *OsSTAR1* and *OsSTAR2* encoded the nucleotide-binding domain and transmembrane region of the ABC transporter, respectively, whilst *OsSTAR1* and *OsSTAR2* masked the binding site of Al^3+^ on the cell wall by regulating the transport of UDP-glucose, thereby improving the Al tolerance of rice [18]. *ALS3* encodes the gene for the ABC transporter and is localized on the cytoplasmic membrane [18]. Larsen et al. [18] found that the main function of *AtALS3* was to transport Al^3+^ from the root tip to the vacuole. The ABA phytohormone pathway is a central regulator of most abiotic stress tolerance mechanisms [20]. Al stress has been found to induce ABA accumulation in various plants [21]. For example, Hou et al. [18] found that abscisic acid (ABA) is involved in the regulation of citrate secretion in soybean roots, whilst the addition of ABA inhibitors inhibits the growth of soybean roots, indicating that ABA induces Al tolerance of soybeans. In addition, previous studies have shown that transcription factors (TFs) such as STOP1, ART1, and WRKY play important roles in plant tolerance to Al [14,22,23].

Peanut (*Arachis hypogaea* L.) is one of the world’s four major oil crops and an essential economic crop in China [15]. Peanut planting areas in Southern China are mainly composed of red and yellow soil having acidic properties. In the second soil survey, the pH of the soil was mostly 6.0–6.5 in South China. Current soil fertility monitoring results show that the pH of the soil has dropped by 0.2–0.5 pH units [24]. According to the physicochemical characteristics of Al in soil, Al can easily be replaced by H^+^ in acidic soils [6]. However, the mechanism of the effect of different concentrations of Al^3+^ on the growth and development of peanut remains less focused. The present study was therefore conducted to assess the effects of low and high concentrations of Al on the growth and transcriptional regulation mechanisms in peanut. This study provides a theoretical basis to understand the ecological adaptability of peanut under Al contaminated conditions.

## 2. Materials and Methods

### 2.1. Plant Materials and Treatment Conditions

The hydroponic experiment was carried out in the Guangzhou Key Laboratory for Research and Development of Crop Germplasm Resources, Zhongkai University of Agriculture and Engineering, Guangzhou, China (23104 N, 113281 E) from March to July 2021. The peanut cultivar “Guihua58” was provided by the Guangxi Academy of Agricultural Sciences, Nanning, China. Homogenous seeds were surface sterilized and germinated in a Petri dish on wet filter paper with 20 seeds per Petri dish. After three days, the seeds were transferred to a plastic culture bowl and continued to cultivate with Hoagland nutrient solution. After one week of growth, peanut seedlings were treated with AlCl_3_.18H_2_O in Hoagland nutrient solution at four different concentrations of Al^3+^, i.e., 0 (pH 6.85), 1.25 (pH 4.03), 2.5 (pH 3.85), and 5 (pH 3.69) mmol/L, and regarded as Al0, Al1, Al2, and Al3. Plant height was measured and recorded every 24 h after exposure to Al treatment. The nutrient solution was replaced once every three days. Leaves were taken after 14 days of treatment. The fresh leaves were stored at −80 °C in a freezer immediately for transcriptome sequencing, ABA content, soluble sugar content, and gene expression determination. Experimental treatments were arranged in a completely randomized design (CRD) with three biological samples and three technical replicates/Data were compiled using Microsoft Excel 2010 (Microsoft, Chicago, IL, USA). SPSS Statistics 20.0 (IBM, Chicago, IL, USA) was used for one-way analysis of variance, and Tukey’s test at the 5% significance level was used to separate treatment means. The R programming language was used for mapping (Auckland, New Zealand).

### 2.2. Raw Sequencing Data

After RNA extraction, magnetic beads with Oligo (dT) were used to enrich eukaryotic mRNA. A fragmentation buffer was added to interrupt mRNA randomly. The first cDNA was synthesized using mRNA as a template and a six-base random hexamer as a primer, then buffer, dNTPs, RNase H, and DNA polymerase I were added to synthesize the second cDNA, and the cDNA was purified with AMPure XP beads. The purified double-stranded cDNA was repaired, A-tailed, and connected to the sequencing adapter, and then AMPure XP beads were used for fragment size selection. Finally, the cDNA library was obtained by PCR enrichment. Qubit 2.0 and Agilent 2100 were used to detect the library’s concentration and insert size, and the library’s effective concentration was accurately quantified using the Q-PCR method to ensure the quality of the library. High-throughput sequencing was performed with NovaSeq 6000, and the sequencing read length was PE150. Initially, the obtained clean reads were filtered to obtain high-quality clean reads. The following filtering criteria were followed: i.e., remove reads containing adapters; remove reads with all A bases; remove reads with N ratio greater than 10%; remove low-quality reads (the number of bases with quality value Q ≤ 20 accounts for more than 50% of the entire reads). The normalization method was fragments per kilobase of transcript per million mapped reads (FPKM) method, and the formula was FPKM = 10^6^C [NL(10^−3^]^−1^ (C is the number of sequenced fragments aligned to gene A, N is the total number of sequenced fragments aligned to the reference gene, and L is the number of bases of gene A). The differentially expressed genes (DEGs) between the comparison groups were obtained by standardizing processing and screening conditions. The default parameters were FDR = 0.01 and FC = 2. Raw sequencing data have been uploaded in the NCBI Gene Expression Omnibus under Accession Number PRJNA754251 (https://www.ncbi.nlm.nih.gov/sra/?term=PRJNA754251 (accessed on 12 August 2021)).

### 2.3. Enrichment Analysis

Raw data analysis was performed using BMKCloud (www.biocloud.net). The differentially expressed genes (DEGs) between the comparison groups were obtained by standardizing processing and screening conditions. The default parameters were FDR = 0.01 and FC = 2. The KEGG database was used for functional annotation, classification statistics, and metabolic pathway analysis of DEGs in the comparison group [15].

### 2.4. Real-Time Quantitative RT-PCR

RNA extraction was performed using the RNAprep Pure Plant Kit (TianGen Biotech, Beijing, China). RNA quality detection was performed using a micro-spectrophotometer (Allsheng, Nano-300, Hangzhou, China). cDNA synthesis used the PrimeScriptRT Reagent Kit with the gDNA Eraser Kit (TaKaRa, Beijing, China). Then, 2× SYBR Green qPCR Mixture Kit (Hlingene Corporation, Shanghai, China) and Option Real-Time PCR System (Bio-Rad, CFX96, Hercules, CA, USA) instruments were used for real-time fluorescent quantitative PCR. Data were analyzed using the 2^−∆∆CT^ method. cDNA was used as a template, three biological replicates were set for each sample, the *actin* gene was used as an internal reference gene, primar 5.0 was used to design qPCR primers, and primers were synthesized by Sangon Biotech (Shanghai, China) (Table 1).

### 2.5. Determination of Soluble Sugar Content

Leaves (50 mg) were ground into powder in an ice bath, and 50 mL of 80% ethanol was used for extraction at room temperature. After centrifuging the supernatant, 80% ethanol was added to make up 10 mL. The extraction was added to a 50 mL centrifuge tube, and 5 mL of anthrone (0.4% in concentrated sulfuric acid) was added. The reaction was stopped after 15 min at 95 °C in a water bath. Sucrose (0.015% *w/v*) was used to construct the standard curve. Soluble sugar content was measured at absorbance at 600 nm [11].

### 2.6. Determination of ABA Content

The cartridge was activated with 4 mL of methanol and 2 mL of 0.1 M aqueous ammonia solution. Fresh leaves (100 mg) were homogenized with 1 mL of extraction solution (acetonitrile/water = 1:1, containing sodium diethyldithiocarbamate) in an ice bath for 4 h at 4 °C and 12,000 rpm for 10 min. The supernatant was concentrated in vacuo; 0.1 M aqueous ammonia solution was added to the volume to 2 mL and then passed through the MAX cartridge. The MAX cartridge was washed with 2 mL of 0.1 M ammonia solution and 2 mL of 0.1 M ammonia solution with 60% methanol, and finally, 0.2 mL of methanol was added to dissolve it. The chromatographic system was an ultra-high-performance liquid system (Vanquish, Thermo, Waltham, MA, USA), and the mass spectrometry system was a Q-executive high-resolution mass spectrometry detection system (Vanquish, Thermo, USA). The liquid chromatography column used was Waters HSS T3 (50 × 2.1 mm, 1.8 μm). The sample volume was 2 μL. The column temperature was 40 °C. The CAS number of the ABA standard product was 14375-45-2, and the gradient concentration of the standard sample was 0.2, 0.5, 1, 10, 40, 80, 120, 160, 200, and 300 ng mL^−1^. The retention time of ABA was 3.58 min. The equation was Y = 4.441 × 10^5^ X. The fit was 0.9992 [15].

## 3. Results

### 3.1. Effects of Different Concentrations of Al^3+^ on the Morphological Characteristics of Peanut and the Expression of AhALMT9 and AhFRDL1 Genes

Compared with Al0, the Al1 treatment increased peanut plant height by 9%. However, Al2 and Al3 reduced plant height by 21% and 40%, respectively (Figure 1A,B). The expression levels of *AhALMT9* and *AhFRDL1* genes were measured in peanut leaves under the different Al concentrations treatment. Compared with Al0, the gene expression of *AhALMT9* and *AhFRDL1* was significantly up-regulated under Al2 and Al3 treatments which were up-regulated by 1.3–1.5 times and 1.7–8.7 times, respectively. Compared with Al0, the *AhALMT9* gene expression was significantly reduced by 77% under the Al1 treatment (Figure 1C). Compared with Al0, the expression of the *AhFRDL1* gene was insignificantly different under the Al1 treatment (Figure 1D).

### 3.2. Sequencing Statistics

Sequencing data of all samples were quality evaluated. The original bases of the sequencing data of each sample were between 5,777,435,056 and 7,275,220,472, the GC content of the sequence was between 45.43% and 46.01%, and the Q30 value was above 92.77% (Table 2).

### 3.3. Mapping of Reads to the Reference Genome

Clean reads were compared with the reference genome, and the number of effective sequences that could be aligned on the reference genome for each sample sequence was between 36,691,674 and 45,441,812, and the contrast ratio was above 94.04% and 8.46% of multiple mapped reads on the reference genome. The unique mapped reads on the reference genome reached more than 84.8% (Table 3). Data showed that the sequencing quality of the transcriptome data was good, the experimental process was pollution-free, and the selected reference genome information could meet the needs of subsequent analysis.

### 3.4. DEGs Identification and KEGG Pathway Analysis

Principal component analysis (PCA) (Figure 2A) and sample-to-sample clustering analysis (Figure 2B) indicated that the same cultivar of control and Al treatment were clustered together. They showed that the library construction quality and sequencing quality results of the sequencing sample were reliable and could be used for subsequent analysis. The distance between each sample point represented the distance of the sample by PCA analysis. The closer the distance, the higher the similarity between the samples, and the better the repeatability between the samples. The Venn diagram showing overlapping and unique DEGs between different peanut samples Al0 vs. Al1, Al0 vs. Al2, and Al0 vs. Al3 is presented in Figure 2C. Through the comparison between samples, the results showed that for Al0 vs. Al1, a total of 633 DEGs were obtained, of which 285 were up-regulated and 348 were down-regulated. For Al0 vs. Al2, 2183 DEGs were obtained, of which 1058 were up-regulated and 1125 were down-regulated. For Al0 vs. Al3, 2294 DEGs were obtained, of which 1447 were up-regulated and 847 were down-regulated. Through the screening and analysis of DEGs between sample groups, it was found that the number of DEGs of Al0 vs. Al3 was the largest, indicating that with the increase in Al concentration, more genes in peanut leaves participated in expression regulation in response to Al stress (Figure 2D).

The DEGs in the Al0 vs. Al1 sample group were enriched in 53 pathways, with 88 genes annotated with KEGG. The DEGs in the Al0 vs. Al2 sample group were enriched into 102 pathways, with 362 genes annotated with KEGG. The DEGs in the Al0 vs. Al3 sample group were enriched in 96 pathways, with 413 genes annotated with KEGG. The KEGG pathway of DEGs was plotted according to the enrichment factor by histogram, and the enrichment results of the top 20 were displayed. The results showed that the KEGG pathways significantly enriched in the Al0 vs. Al1 sample group were starch and sucrose metabolism pathway. The KEGG pathways were significantly enriched in the Al0 vs. Al2 and Al0 vs. Al3, including the phenylpropanoid biosynthesis and plant hormone signal transduction pathway (Figure 3). Therefore, low-concentration Al1 treatment focused on starch and sucrose metabolism pathway, and high-concentration Al2 and Al3 treatments focused on plant hormone signal transduction pathway.

### 3.5. Effect of Different Al^3+^ Concentrations on the Soluble Sugar Content and AhERD6 Transcript Levels

The content of soluble sugar in peanut leaves was determined under different Al concentrations. Compared with Al0, the soluble sugar content was increased by 2.86 times under Al1 treatment, but found statistically to with Al2 and Al3 treatments (Figure 4A). The expression of the *AhERD6* gene in peanut leaves was determined under different Al concentrations. Compared with Al0, the expression of *AhERD6* increased by 2.22 times in the Al1 treatment, there was no significant difference in the Al2 treatment, and it decreased by 40% in the Al3 treatment (Figure 4B).

### 3.6. Effect of Different Al^3+^ Concentrations on the ABA Content and AhNCED1 Transcript Levels

The content of ABA in peanut leaves was determined under different Al concentrations. Compared with Al0, the ABA content was increased by 59% and 3.02 times under the Al2 and Al3 treatments, respectively, but found statistically similar to the Al1 treatment (Figure 5A). NCED was the key enzyme involved in ABA synthesis in higher plants [25]. In addition, the expression of *AhNCED1* increased by 4.91 and 5.52 times in Al2 and Al3 treatments, respectively, compared with Al0. Moreover, no significant difference was noted for the Al1 treatment regarding *AhNCED1* (Figure 5B).

### 3.7. Effect of Different Al^3+^ Concentrations on the Regulation of TFs Expression

The key TFs associated with Al^3+^ treatments in peanut were exhibited in Figure 5. Among them, the MYB (4), bHLH (9), NAC (6), ASR (7), STOP1 (3), MADS-box (5), ABI5 (2), RAE (4), and WRKY (15) were differently expressed in different treatments (Figure 6A). Ten genes from the differentially expressed transcription factor genes were randomly selected for validation by qRT-PCR analysis (Figure 6B). The randomly selected gene expression was consistent with the results calculated by the FPKM value obtained by sequencing, indicating that the transcriptome data were reliable (Table 4).

## 4. Discussion

Heavy metal in excess concentrations often results in deleterious impacts on crop plants [26,27]. Among the plethora of heavy metals, Pb, Cd, As, Cr, and Al are amongst the most dangerous for plant growth and productivity and are widely reported in metal-polluted soils [28,29,30,31]. The study found that low concentrations of Al1 treatment promoted plant height in peanut, while high concentrations of Al2 and Al3 inhibited plant height. The expression of the *AhALMT9* and *AhFRDL1* genes was up-regulated under high concentrations of Al^3+^, i.e., Al2 and Al3, whereas low concentrations of Al1 treatment did not promote the expression of these two genes. STOP1 is a key transcription factor that positively regulates the expression of *ALMT1* in plant aluminum tolerance [32]. RAE1 encoded an F-box protein capable of degrading STOP1 by ubiquitination [33]. Therefore, according to previous studies, we speculated that under high concentrations of Al^3+^, i.e., Al2 and Al3, the STOP1 protein was accumulated, and the expression of RAE1 was induced by STOP1 at the same time. RAE1 degrades the excessive accumulation of the STOP1 protein through ubiquitination and promotes the expression of the *AhALMT9* gene. Under the action of low concentrations of Al1, the accumulation of STOP1 was not affected, so the *AhALMT9* gene expression induced by STOP1 was insignificant. Similarly, for the *AhFRDL1* gene, there was similar regulation at the transcriptional level. 

This study showed that the DEGs in the Al0 vs. Al1 sample group were significantly enriched in starch and sucrose metabolic pathways. The DEGs in the Al0 vs. Al2 and the Al0 vs. Al3 sample groups were significantly enriched in the plant hormone signal transduction pathway (Appendix A Appendix A). Soluble sugars not only played their roles in cellular metabolism and structural components, but also served as signals regulating plant growth and development [11,34,35]. Previous studies have found that the soluble sugar concentration may be closely correlated with tolerance to saline stress [36]. Low concentrations of Al^3+^ significantly increased the soluble sugar content in rice shoots [11]. It was found in sunflower that Al^3+^ treatments at 100 μM and 200 μM significantly increased the soluble sugar content [37]. These results were different from Hajiboland et al. [38], who reported that the soluble sugar contents in young leaves and roots of tea were not affected substantially under Al^3+^ treatment, whereas the soluble sugar content in old leaves was decreased. However, this study found that low concentrations of Al1 promoted peanut growth by increasing the content of soluble sugar, and the content of soluble sugar after high concentrations of Al2 and Al3 treatments was statistically similar to the control, indicating that low concentrations of Al1 could stimulate peanut growth and increase biomass). ABA induced the expression of *AtALMT1*, suggesting that ABA acted as a signal involved in the mechanism of Al tolerance [39]. Exogenous ABA enhanced Al-induced secretion of soybean citrate [18]. *NCED* involved in ABA biosynthesis was induced by Al stress and regulated stomatal conductance [40]. After treatment of rice root tips with ABA and Al, respectively, one-third of differentially expressed genes were found to be identical, suggesting potential crosstalk between Al and ABA signaling [41]. This study found that high concentrations of Al2 and Al3 promoted the up-regulation of ABA content in peanut leaves, while low concentrations of Al1 did not increase ABA content, indicating that high concentrations of Al2 and Al3 made peanuts sensitive to Al, induced the expression of *AhNCED1*, and increased endogenous ABA content.

In the present study, it was found that the expression of a large number of TFs was different in different concentrations of Al^3+^ treatment in peanut. Genes encoding bHLH, NAC, and STOP1 have negative FC in Al1 and gradually become positive FC in Al2 and Al3. WRKY46 could negatively regulate the expression of *ALMT1* under Al stress, but WRKY22 could positively regulate the expression of *FRDL4* under Al stress in rice [42]. The expression of ASR was induced by ABA, and the expression of the *ASR1* and *ASR5* genes was also induced by Al stress [43]. *ASR1* and *ASR5* have complementary roles, specifically binding to a cis-acting element in the *STAR1* (sensitive to Al rhizotoxicity) promoter to activate *STAR1* expression [44]. *STAR1* and *STAR2* encoded ATP-binding proteins that transported UDP-glucose and masked Al-binding sites in the cell wall [44]. GsMAS1 was a MADS-box transcription factor, and overexpression of *GsMAS1* in *Arabidopsis* could improve Al tolerance [45]. Al treatment induced the expression of the NAC gene in rice, and Al promoted its growth by mediating the expression of NAC [46]. ABI5 positively regulated tolerance against Al stress in rice [41]. STOP1 and ART1 could positively regulate the expression of genes responsive to Al stress (*FRDL* and *ALMT*, etc.), but the genes encoding STOP1 and ART1 did not respond to Al-induced expression [47]. The zinc finger transcription factor STOP1 was a key transcription factor regulating the expression of *AtALMT1*, which may be regulated by Al at the post-transcriptional or post-translational level [23,32,33]. ART1 and STOP1 play a very important role in regulating Al tolerance in plants. Studies have found that after the ART1 and STOP1 fragments in tobacco (*Nicotiana tabacum*) and moss (*Physocmitrella patens*) are knocked out, plants are observed to be sensitive to Al^3+^ [18]. Therefore, The TFs played an important role in Al regulation of peanut growth and development.

## 5. Conclusions

Low concentrations of Al1 (1.25 mmol/L pH 4.03) promoted peanut growth by increasing soluble sugar content, while high concentrations of Al2 (2.5 mmol/L pH 3.85) and Al3 (5 mmol/L pH 3.65) inhibited peanut growth and induced the production of endogenous ABA content. Therefore, this study showed that Al acted as a stimulant to promote the accumulation of peanut biomass. However, the adaptive mechanism of peanut in response to Al stress needs further research in the future.

## Figures and Tables

**Figure 1 genes-13-01830-f001:**
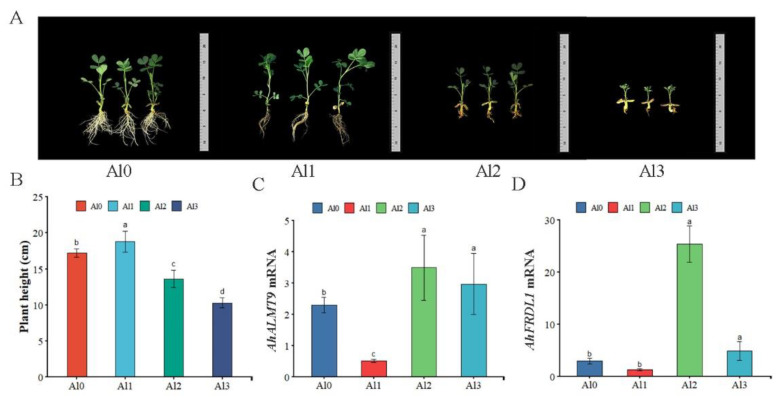
Effects of different concentrations of Al^3+^ on the morphological characteristics of peanut and the expression of *AhALMT9* and *AhFRDL1* genes. Plant height was measured for 14 days of Al treatment (**A**). Effect of different concentrations of Al on peanut plant height (**B**). qRT-PCR analysis of relative transcription levels of *A**hALMT9* (**C**) and *Ah**FRDL1* (**D**). Al0, Al1, Al2 and Al3 represent 0 (pH 6.85), 1.25 (pH 4.03), 2.5 (pH 3.85) and 5 (pH 3.69) mmol/L AlCl_3_.18H_2_O solutions, respectively. Data are means (±SE), *n* = 3. Marking the same letters means *p* ≥ 0.05 (LSD), there is no significant difference, the difference between different letters means *p* < 0.05 (LSD), and the difference is significant.

**Figure 2 genes-13-01830-f002:**
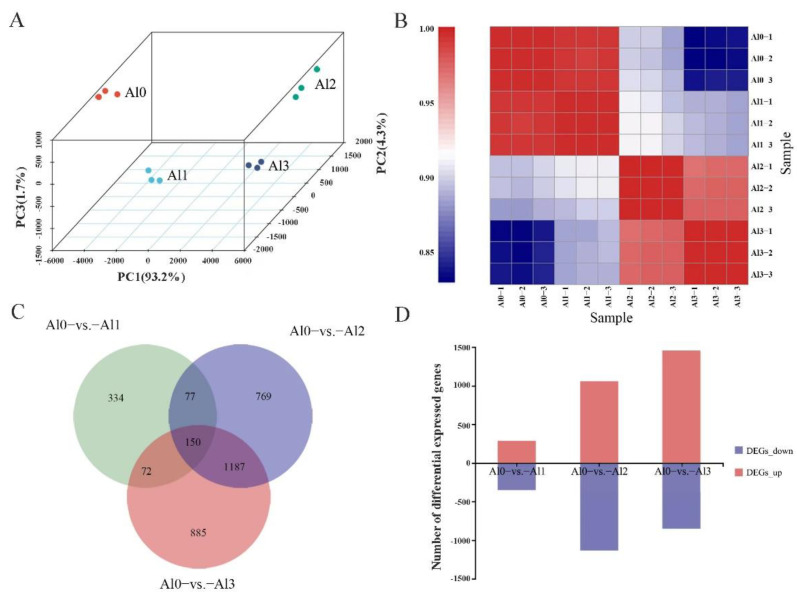
DEG identification and KEGG pathway analysis. Principal component analysis (PCA). The same color represents the same treatment, different colors represent different treatments (**A**). Sample-to-sample clustering analysis. The red color depicts the up-regulation, and blue color depicts the down-regulation in the gene expression pattern (**B**). Venn diagram showing overlapping and unique DEGs between different peanut samples Al0 vs. Al1, Al0 vs. Al2, and Al0 vs. Al3 (**C**). Statistical chart of DEGs under different concentrations of Al^3+^. Red indicates up-regulation, and blue indicates down-regulation (**D**).

**Figure 3 genes-13-01830-f003:**
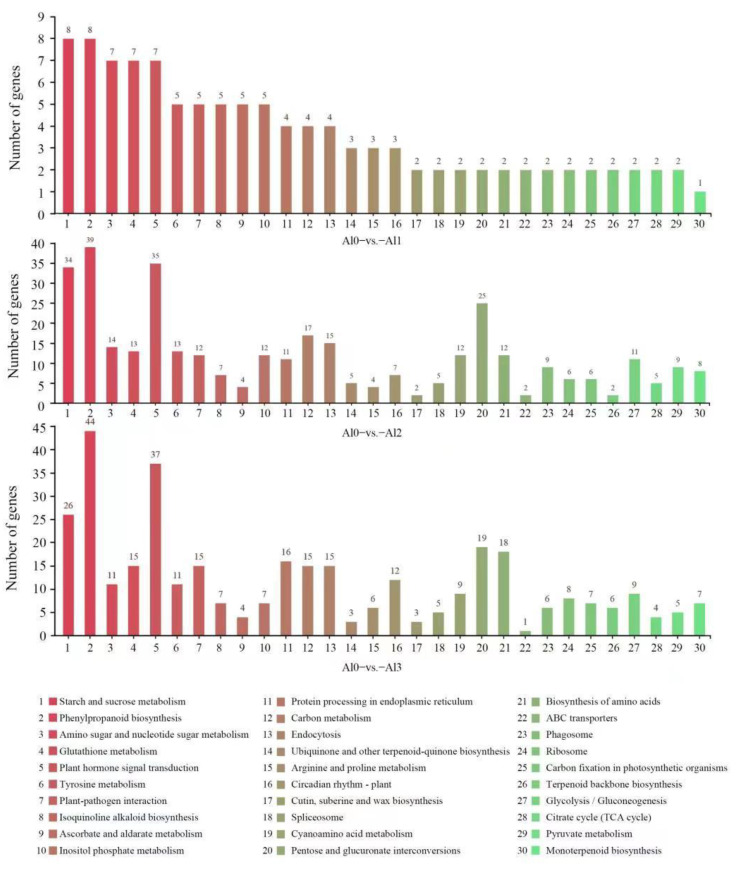
Regulations of the KEGG pathway. Al0, Al1, Al2, and Al3 represent 0 (pH 6.85), 1.25 (pH 4.03), 2.5 (pH 3.85), and 5 (pH 3.69) mmol/L AlCl_3_.18H_2_O solutions, respectively. Different colors represent different biological processes, and numbers represent the number of DEGs in the same biological process under different Al treatment conditions.

**Figure 4 genes-13-01830-f004:**
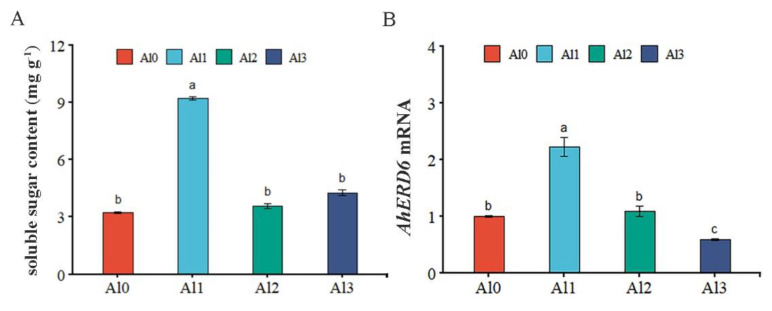
Effect of different Al^3+^ concentrations on the soluble sugar content and *AhERD6* transcript levels. The effect of Al treatments on soluble sugar content (**A**). qRT-PCR analysis of relative transcription levels of *AhERD6* (**B**). Al0, Al1, Al2, and Al3 represent 0 (pH 6.85), 1.25 (pH 4.03), 2.5 (pH 3.85), and 5 (pH 3.69) mmol/L AlCl_3_.18H_2_O solutions, respectively. Data are means (±SE), *n* = 3. Marking the same letters means *p* ≥ 0.05 (LSD), there is no significant difference, the difference between different letters means *p* < 0.05 (LSD), and the difference is significant.

**Figure 5 genes-13-01830-f005:**
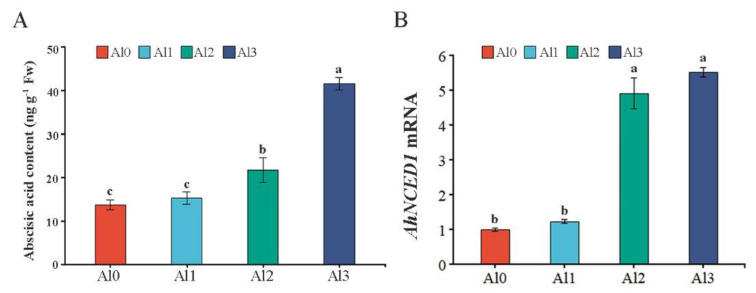
Effect of different Al^3+^ concentrations on the ABA content and *AhNCED1* transcript levels. The effect of Al treatments on ABA content (**A**). qRT-PCR analysis of relative transcription levels of *AhNCED1* (**B**). Al0, Al1, Al2, and Al3 represent 0 (pH 6.85), 1.25 (pH 4.03), 2.5 (pH 3.85), and 5 (pH 3.69) mmol/L AlCl_3_.18H_2_O solutions, respectively. Data are means (±SE), *n* = 3. Marking the same letters means *p* ≥ 0.05 (LSD), there is no significant difference, the difference between different letters means *p* < 0.05 (LSD), and the difference is significant.

**Figure 6 genes-13-01830-f006:**
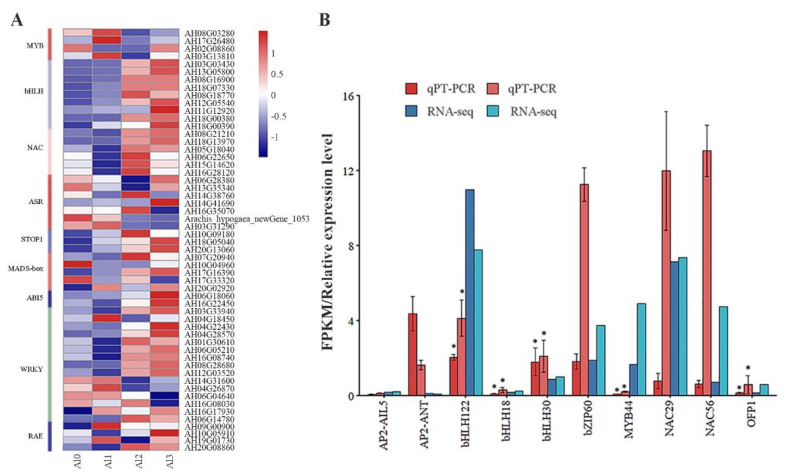
Effect of different Al^3+^ concentrations on the regulation of TFs expression. Heatmap of known 38 TFs genes. Red means down-regulation, and blue means up-regulation (**A**). Compare the FPKM value obtained by RNA-seq analysis with the gene relative transcript levels obtained by qRT-PCR analysis (**B**). * means *p* ≥ 0.05 (LSD). Data were analyzed using the 2^−∆∆CT^ method.

**Table 1 genes-13-01830-t001:** Primer sequences of genes.

Gene Name	Gene ID	Primer Sequences
*AhFRDL1*	*LOC112769177*	F 5′-TGATGCTGAAACCAAAGAGTTCCTACC-3′
		R 5′-CAGATGAAGCCGAAGGGATATGCC-3′
*AhALMT9*	*LOC112802199*	F 5′-ACCACACTCTCTCCCTCCAAATCC-3′
		R 5′-ACAGCAGCCTCTCCTTCTTCTCC-3′
*AhWRKY1*	*LOC112766106*	F 5′-TTGCTTGGATACTTGGATTGGACTCTC-3′
		R 5′-TGGTTGGTTGGTTGGTTGAATTGAATG-3′
*AhWRKY23*	*LOC112710566*	F 5′-AAGCGTGTGGAGCGGTCATTTAC-3′
		R 5′-AATGGCTGAACGAGGCATAACTGG-3′
*AhWRKY44*	*LOC112759075*	F 5′-ACAGTCACAGCATTACACCACCTTC-3′
		R 5′-AGCCTTCTGATCCTCCTCTATGTTCTG-3′
*AhOFP1*	*LOC112728150*	F 5′-ACCCCAAAGATTCACCACCACAAAG-3′
		R 5′-AGAAGAGACAGAGGAGGAGGAAGTAAC-3′
*AhOFP13*	*LOC112780116*	F 5′-GGAGATGGTTGAGTGTCATGGAGTG-3′
		R 5′-GAAGCAGAAGCAGCAATGGTGATAATC-3′
*AhbZIP60*	*LOC112727401*	F 5′-AACCATGCTTTGCGTCTTTGCTTAC-3′
		R 5′-AGTCAATGTGTTACTGTAGGGCTTACC-3′
*AhAP2-AIL5*	*LOC112723564*	F 5′-TGGCAATGGCATGATGGACTTCTC-3′
		R 5′-ACCACCACTCTCTTGTTGATGATGATG-3′
*AhAP2-ANT*	*LOC112756010*	F 5′-CTTCAAGCACTGCCTCAACAACAAG-3′
		R 5′-GCCAGAGAAGGAGAAGAATGAAAGGG-3′
*AhNAC29*	*LOC112759194*	F 5′-TGGTCATGCCGTTGCGTTCTAC-3′
		R 5′-ACCACCCTCACTCACATCGTCTC-3′
*AhNAC56*	*LOC112707370*	F 5′-TTCTGGAAGGTATGATGCTAGGAGGAG-3′
		R 5′-TGTTAATGGTGTTTGAGGCTGATCCC-3′
*AhbHLH13*	*LOC112758465*	F 5′-TGCTGTTAGGAAGATGGAGGAGAGG-3′
		R 5′-GATGAAGACCATTGCGAGGAGGAC-3′
*AhbHLH18*	*LOC112720570*	F 5′-GCCACCACCAGAAACCCAAC-3′
		R 5′-AGTCCAGGAACAATGGCGGA-3′
*AhbHLH30*	*LOC112802694*	F 5′-TCACGGCGGACGAAGAAGATTATTC-3′
		R 5′-AGCAGATTCATCACCACCACCTTTC-3′
*AhbHLH96*	*LOC112765920*	F 5′-GGACATAGAAGTGACAATGGTGGACAG-3′
		R 5′-AGGTGGAGAATGGTGAGGCTTAGAG-3′
*AhbHLH115*	*LOC112772203*	F 5′-GGACATAGATTCTTCAGGAGGCACATC-3′
		R 5′-CAGCCAGGTGAAAGCAGCAGAG-3′
*AhbHLH122*	*LOC112706635*	F 5′-ATGTTGTTGATATGCGAGGGATGGG-3′
		R 5′-GAGGCGGAGGAGGAGTAGTTCTG-3′
*AhMYB14*	*LOC112722417*	F 5′-AACCTGCACCGTGTCTGATGTTG-3′
		R 5′-AAGCCGAATGTCTTGATGGAGTCTG-3′
*AhMYB44*	*LOC112766377*	F 5′-GACCCGTTGACTGCGTTGACTC-3′
		R 5′-CTCACTTCCCTGGCAATCACATCC-3′
*AhERD6*	*LOC112741105*	F 5′-ACAGCATTGGGAGCAATACTGATGG-3′
		R 5′-CAGCCTACGAAAGTGCCACTAGC-3′
*AhNCED1*	*LOC112703460*	F 5′-TGGAAGGAAGACACAGTTCGCATAC-3′
		R 5′-CTTCGCCGCTGGACAGATCAAC-3′
*Ahactin*	*LOC112787680*	F 5′-AAGCTGGCTTACATTGCCCT-3′
		R 5′-TGACCTGTCCATCAGGCAAC-3′

**Table 2 genes-13-01830-t002:** Sequencing statistics.

Sample	Clean Reads	Clean Bases	GC Content	% ≥ Q30	SRX
Al0-1	20,553,687	6,146,509,322	46.01%	93.00%	11,763,364
Al0-2	23,425,837	7,009,817,148	45.85%	92.97%	11,763,363
Al0-3	24,343,990	7,275,220,472	45.96%	93.16%	11,763,362
Al1-1	23,330,520	6,975,644,336	45.84%	93.14%	11,763,361
Al1-2	22,315,847	6,672,118,448	45.85%	92.97%	11,763,360
Al1-3	20,976,609	6,272,951,552	45.98%	92.96%	11,763,359
Al2-1	19,314,435	5,777,435,056	45.50%	92.89%	11,763,358
Al2-2	21,413,255	6,403,612,920	45.53%	93.07%	11,763,357
Al2-3	21,233,487	6,346,252,156	45.43%	92.81%	11,763,356
Al3-1	21,037,988	6,290,647,134	45.44%	92.81%	11,763,355
Al3-2	22,477,591	6,720,640,752	45.43%	92.77%	11,763,354
Al3-3	22,844,859	6,834,289,998	45.49%	92.88%	11,763,353

Note: Q30 values represent the number and percentage of bases with sequencing accuracy of 99.9%.

**Table 3 genes-13-01830-t003:** The mapping of reads to the reference genome.

Sample	Total Reads	Mapped Reads	Unique Mapped Reads	Multiple Map Reads
Al0-1	41,107,374	38,674,767 (94.08%)	35,195,591 (85.62%)	3,479,176 (8.46%)
Al0-2	46,851,674	44,198,905 (94.34%)	40,263,962 (85.94%)	3,934,943 (8.40%)
Al0-3	48,687,980	45,441,812 (93.33%)	41,288,361 (84.80%)	4,153,451 (8.53%)
Al1-1	46,661,040	43,883,594 (94.05%)	39,896,921 (85.50%)	3,986,673 (8.54%)
Al1-2	44,631,694	42,028,809 (94.17%)	38,235,448 (85.67%)	3,793,361 (8.50%)
Al1-3	41,953,218	39,454,492 (94.04%)	35,906,710 (85.59%)	3,547,782 (8.46%)
Al2-1	38,628,870	36,691,674 (94.99%)	33,371,783 (86.39%)	3,319,891 (8.59%)
Al2-2	42,826,510	40,650,589 (94.92%)	36,919,727 (86.21%)	3,730,862 (8.71%)
Al2-3	42,466,974	40,121,437 (94.48%)	36,392,658 (85.70%)	3,728,779 (8.78%)
Al3-1	42,075,976	39,782,778 (94.55%)	36,169,774 (85.96%)	3,613,004 (8.59%)
Al3-2	44,955,182	42,699,985 (94.98%)	38,830,480 (86.38%)	3,869,505 (8.61%)
Al3-3	45,689,718	43,437,766 (95.07%)	39,532,547 (86.52%)	3,905,219 (8.55%)

**Table 4 genes-13-01830-t004:** FC values calculated from randomly selected gene expression and FPKM values obtained by sequencing.

Gene	Sample	FC	Gene	Sample	FC
*AhOFP1*	Al0_RNA-seq		*AhOFP1*	Al0_qRT-PCR	
*AhOFP1*	Al1_RNA-seq	4.2249179	*AhOFP1*	Al1_qRT-PCR	4.4242215
*AhbZIP60*	Al0_RNA-seq		*AhbZIP60*	Al0_qRT-PCR	
*AhbZIP60*	Al1_RNA-seq	1.9782843	*AhbZIP60*	Al1_qRT-PCR	6.1841118
*AhbHlH122*	Al0_RNA-seq		*AhbHlH122*	Al0_qRT-PCR	
*AhbHlH122*	Al1_RNA-seq	0.7069601	*AhbHlH122*	Al1_qRT-PCR	2.0390607
*AhbHlH18*	Al0_RNA-seq		*AhbHlH18*	Al0_qRT-PCR	
*AhbHlH18*	Al1_RNA-seq	1.4979671	*AhbHlH18*	Al1_qRT-PCR	3.4426249
*AhMYB44*	Al0_RNA-seq		*AhMYB44*	Al0_qRT-PCR	
*AhMYB44*	Al1_RNA-seq	2.9794002	*AhMYB44*	Al1_qRT-PCR	2.4741158
*AhAP2-AIl5*	Al0_RNA-seq		*AhAP2-AIl5*	Al0_qRT-PCR	
*AhAP2-AIl5*	Al1_RNA-seq	1.1578537	*AhAP2-AIl5*	Al1_qRT-PCR	2.1525224
*AhAP2-ANT*	Al0_RNA-seq		*AhAP2-ANT*	Al0_qRT-PCR	
*AhAP2-ANT*	Al1_RNA-seq	0.8790023	*AhAP2-ANT*	Al1_qRT-PCR	0.3706083
*AhbHlH30*	Al0_RNA-seq		*AhbHlH30*	Al0_qRT-PCR	
*AhbHlH30*	Al1_RNA-seq	1.1577513	*AhbHlH30*	Al1_qRT-PCR	1.1665719
*AhNAC56*	Al0_RNA-seq		*AhNAC56*	Al0_qRT-PCR	
*AhNAC56*	Al1_RNA-seq	6.5449812	*AhNAC56*	Al1_qRT-PCR	20.860103
*AhNAC29*	Al0_RNA-seq		*AhNAC29*	Al0_qRT-PCR	
*AhNAC29*	Al1_RNA-seq	1.0325376	*AhNAC29*	Al1_qRT-PCR	15.497673

## Data Availability

The authors affirm that all data necessary for confirming the conclusions of the article are present within the article, figures, and tables.

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
