# Peer review of "Transcriptomic Analysis Provides Insights into the Differential Effects of Aluminum on Peanut (Arachis hypogaea L.)"

_genes, 2022, doi:10.3390/genes13101830_

Round 1
Reviewer 1 Report
Comments for the Authors
In the manuscript, authors have given an exhaustive account of the “Transcriptomic analysis provides insights into the differential 2 effects of aluminum on peanut (Arachis hypogaea L.)”. The authors have put in good number of efforts, but the paper suffers from poor English and too much grammatical mistakes throughout the manuscript. Overall, I consider that the paper adopts good contribution to this field of research but still need careful consideration. I recommend this article to accept for publication after minor revision. For the better readability of the manuscript, I would like to recommend changes and improvements as follow:
1. Name of the genes and strains should be italic.
2. There are several mistakes in the whole manuscript, please add or remove extra characters, and check again.
3. Please improve the introduction and abstract part of the paper.
4. Conclusion section is also not well written please improved it.
5. English should be carefully checked and improved. The paper suffers from poor grammatical sentences and mistakes.
6. Please carefully check the reference style of this journal.
7. Please thoroughly read the overall paper.
Author Response
In the manuscript, authors have given an exhaustive account of the “Transcriptomic analysis provides insights into the differential effects of aluminum on peanut (Arachis hypogaea L.)”. The authors have put in good number of efforts, but the paper suffers from poor English and too much grammatical mistakes throughout the manuscript. Overall, I consider that the paper adopts good contribution to this field of research but still need careful consideration. I recommend this article to accept for publication after minor revision. For the better readability of the manuscript, I would like to recommend changes and improvements as follow:
Thank you for reading our manuscript and reviewing it, which will help us improve it to a better scientific level. We revised our manuscript, and quite a lot of changes have taken place. So, we have sent the revised manuscript, and a version containing all the changes to be visible. At the following, the points mentioned by the reviewers will be discussed:
- Name of the genes and strains should be italic.
Response 1:We checked all gene and strain names, which have been modified to italics.
- There are several mistakes in the whole manuscript, please add or remove extra characters, and check again.
Response 2:Spelling mistakes have been corrected.
- Please improve the introduction and abstract part of the paper.
Response 3: Introduction and abstract sections have been refined.
- Conclusion section is also not well written please improved it.
Response 4: Conclusion has been revised.
- English should be carefully checked and improved. The paper suffers from poor grammatical sentences and mistakes.
Response 5: We invited a native English-speaking Ph.D. to help polish our article. And we hope the revised manuscript could be acceptable for you.
- Please carefully check the reference style of this journal.
Response 6: We carefully review the journal's referencing style and follow the journal's format.
- Please thoroughly read the overall paper.
Response 7:Yes, the full text has been carefully checked.

Reviewer 2 Report
The authors have presented results about Aluminium toxicity in Peanut. Overall it’s a good piece of work and data. However, there are some strong loopholes in data representation, results to be done, and writing. Materials, methods and figure legends are very poorly written, needs considerable improvement. Figure legends should be enough explanatory to understand without much search in the paper context. I recommend authors to consider each point raised very carefully and revise the MS with more data, with one table of gene expression of selected genes from transcriptome sequencing. The authors are encouraged to resubmit the MS with more data after revision
Abstract:
No need to mention the year in the abstract. Overall there is a need of better representation. pH are twice mentioned, once at the beginning and once at the end of abstract. No mention about the genes expression. Interpret one line about the mechanism of tolerance and susceptibility from the result obtained in the paper.
Introduction:
Line 42: H+
Line 57-59, Rephrase, not understandable
Programmed cell death (PCD) … of its exhibits
Line 69: Al3+
line 68: rephrase, grammar incorrect
The internal tolerance mechanism is that…
Line 76: rephrase, grammar incorrect
The expression of AhFRDL1 gene was up-regulated and increased…
line 79: Directly changed to ABA phytohormone. Connecting evidence of Aluminium stress in A.hypogea with other hormones too among which ABA should be major.
The ABA signaling…
line 119: Fertility
The Introduction finds itself elaborative. However, it is not systematically structured for the context of the research work done in this paper. It should address
1. Significance of Aluminimum toxicity in terms of crop growth/ yield, symptoms in plant etc.
2. Complicacy of aluminimum forms in terms of acidic/basic, high / low concentration, interaction with phosphorus, iron etc.
3. Signaling of aluminium starting from uptake till growth response, genes known, explaining networks with hormone signaling, any genomic work done
4. Gaps to be addressed in this paper and its significance
Eliminate irrelevant and not so related topics like, other crop tea, peony, alfalfa, batteries, buck wheat etc.
Eliminate unnecessary contexts like definition of PCD. Make clear and short sentences, short context on: in which form aluminium is taken up by plant.
M&M
Experiment time frame was only 25 days. However, how many biological replicates are done to generate the samples?
Why morphology (Figure 1) was done in 2 weeks, whereas transcriptome, gene expression was done in 25 days?
Reviewers link for the NCBI data submission is missing.
Experimental details of probes found, DEGs obtained, normalization, filtering criteria etc. missing
Calculation of Fold change procedure for the QRT PCR missing
Cartridge details with model no, Detection of ABA, retention time, standards used etc are missing for ABA quantification
Results:
Line 216: not significantly different: replace with : insignificant
Figure 1 Legends: A, B: 15 day old plants, C,D: 25 day old plants. Not written clearly in legends. Is this absolute transcription level or relative transcription level? Or Log 2 Fold change?
Line226: Q20 value is not mentioned in Table 1!!!, but mentioned to be present in Table 1. Significance of the Q20 and Q30 values need to be mentioned
A statistics of similarity of the replicates from same treatment and dis-similarity of replicates of different treatments is highly required before genomic analysis, it could be shown as higher correlation among the replicates of same treatment and less correlation among data of other treatments.
Line 249: Figure S1 is missing but mentioned in text. Figure 2 Legends is completely missing
Figure 2: Y-axis labels Number of gens missing. A venn-diagram showing DEGs among different treatment sets would be clearer.
Colour codes are creating confusion due to being similar colour, each histogram bar could be numbered instead. The pathway classification is arranged according to Al 0 Vs Al 1. However, pathway classification of the rest two treatments should also be taken individually not to skip any significant observation in them.
Line 267, line 270: ….increased by 186.91 % etc. It is better to calculate in how many folds in this case
Section 3.6: similarly anything above 100% could be better explained by fold level.
A.hypogea AhNCED1 is involved ABA signaling. However, which NCED here is not mentioned.
All figure legends are not explanatory. It should be elaborated with little sample and experiment description to understand the figure and correlate with the result text.
Line No 296: consistency of the FC in the genomic data and QPCR data must be shown in table and not mentioned in one line text. In Figure 5B, in addition to the random selection of genes, the Al signalling genes like, ASR, STOP, RAE etc QRTPCR could strengthen the claim the results of this paper
Discussion:
The 1st paragraph explains about mechanisms of Al stress tolerance at molecular level, it can be preferable shifted to Intro. Even the 2nd paragraph, line no 329-332: significance of transcriptome bears its relevant place in intro
No need to mention figure numbers again in discussion.
line 303: …..is root production organic acids….: correct grammar as
production of organic acids by roots
line 310, Line 368: Arabidopsis : capital and italics
….which have been confirmed…. Replace with correct grammar sentence
line 316-328: There is a great problem in explanation of the data.
STOP1 expression is shown in Figure 5A. This shows STOP1 is negative FC in Al1 and gradually becomes positive FC in Al2 and Al3. So -ve STOP1 could not induce the ALMT9 in Al1. And why STOP1 and RAE1 QRTPCR is not shown in Figure 5B? Again FDL1 regulation at protein level is not shown in Figure 1D, why there is a mention of protein level here?
Line No 333-335: Grammar incorrect, rephrase the sentence. The authors want to say that Al2, 3 are having induced with ABA pathway which is not found in Al1. In that case the pathway analysis for all 8 hormones should be shown in A1,2, 3. In Figure 5A, fold expression FPKM value of all TFs bears no relevance here, rather all the hormone signalling and the Al signalling genes’ heat map would have more claim on data.
Wrong mention of Reference 46: line No: 348
ABA induced the expression of AtALMT1….. is shown in Kobayashi et al. 2013, Reference Number 43.
Line 348-352: the data and the statements are not enough to claim that “ABA improved Al tolerance by modulating organic acids.” Again Reference 47 doesn’t explain “ABRE existed in the VUMATE promoter region”. It is a wrong mention of reference. Together there is not enough data to claim higher ABA content and gene expression regulated organic acid secretion. For this claim, more data required to show in terms of organic acid secretion.
Line 355-358: great confusion in explaining data.
If Al2,3 show Aluminium tolerance by increasing ABA content, then why it is not visible in morphology of the Al2,3? So increase in ABA content and NCED expression is for Al tolerance phenotype or for sensitive phenotype?
Line 364: again there is wrong mention of Referenc 51, I think the ASR1 and ASR5 reference is present in 47.
I think the authors should carefully correlate the whole reference list with the text and renumber them again.
Line 359-376: This paragraph discusses basically Figure 5A, heatmap, fold expression of TFs. It should be actually discussed in the result section first. Moreover, In the discussion part, along with other’s references, the present data should be discussed.
Author Response
The authors have presented results about Aluminium toxicity in Peanut. Overall it’s a good piece of work and data. However, there are some strong loopholes in data representation, results to be done, and writing. Materials, methods and figure legends are very poorly written, needs considerable improvement. Figure legends should be enough explanatory to understand without much search in the paper context. I recommend authors to consider each point raised very carefully and revise the MS with more data, with one table of gene expression of selected genes from transcriptome sequencing. The authors are encouraged to resubmit the MS with more data after revision
We would like to thank the reviewer for careful and thorough reading of this manuscript and for the thoughtful comments and constructive suggestions, which help to improve the quality of this manuscript. Our response follows:
Abstract:
Point 1:No need to mention the year in the abstract. Overall there is a need of better representation. pH are twice mentioned, once at the beginning and once at the end of abstract. No mention about the genes expression. Interpret one line about the mechanism of tolerance and susceptibility from the result obtained in the paper.
Response 1:Line 14-34 Abstract has been revised.
Abstract: In acidic soils, high concentrations of aluminum ions (Al3+) in dissolved form reduces root growth and development of most crops. In addition, Al3+ is also a beneficial element in some plant species in low concentrations. However, the regulatory mechanism of the growth and development of peanut (Arachis hypogaea L.) treated with different concentrations of Al3+ has been rarely studied. In this study, peanut seedlings were treated with AlCl3.18H2O in Hoagland nutrient solution at four different concentrations of Al3+ i.e., 0 (pH 6.85), 1.25 (pH 4.03), 2.5 (pH 3.85) and 5 (pH 3.69). mmol l-1, which are regarded as Al0, Al1, Al2 and Al3. The results showed that low-concentration of Al treatment (Al1) promoted peanut growth, while high-concentration of Al treatments (Al2 and Al3) significantly inhibited the peanut growth. Compared with the control (Al0), transcriptome analysis showed that the differentially expressed genes (DEGs) of starch and sucrose metabolic pathways were significantly enriched at low-concentration i.e., Al1 treatment, whereas the expression of ERD6 was significantly up-regulated, and the soluble sugar content was significantly increased; The DEGs of the plant hormone signaling transduction pathway were significantly enriched at high concentrations of Al2 and Al3 treatments, whereas the expression of NCED1 was significantly up-regulated, and the content of ABA was significantly increased. Moreover, the expression of transcription factors (TFs) in peanut were affected by different concentrations of Al. Overall, low concentration of Al1 promoted peanut growth by increasing soluble sugar content, while high concentrations of Al2 and Al3 inhibited the growth of peanut, and induce the NCED1 gene expression and increase the endogenous ABA content. For peanut, the exposure of Al at low concentration not only derived an adaptive mechanism to cope with Al stress, but also acted as a stimulator to promote its growth and development.
Introduction:
Point 1:Line 42: H+
Response 1:Corrected H+ to H+. We feel sorry for our carelessness. In our resubmitted manuscript, the typo is revised. Thanks for your correction.
Point 2:Line 57-59, Rephrase, not understandable
Programmed cell death (PCD) … of its exhibits
Response 2:The definition of programmed cell death (PCD) has been removed.
Point 3:Line 69: Al3+
Response 3:Corrected Al3+ to Al3+.
Point 4:line 68: rephrase, grammar incorrect
The internal tolerance mechanism is that…
Response 4:Line 71-73. This sentence has been changed to “The internal tolerance mechanism involves the adsorption of Al3+ by hemicellulose and pectin in the cell wall and the transfer of Al3+ into the vacuole by organic acid transporters in the cytoplasm.”
Point 5:Line 76: rephrase, grammar incorrect
The expression of AhFRDL1 gene was up-regulated and increased…
Response 5:Line 77-79. This sentence has been changed to “The Al induced the expression of AhFRDL1 gene and the secretion of citrate in peanut root tips, and the AhFRDL1 was involved in peanut aluminum tolerance by regulating citrate secretion.”
Point 6:line 79: Directly changed to ABA phytohormone. Connecting evidence of Aluminium stress in A.hypogea with other hormones too among which ABA should be major.
The ABA signaling…
Response 6:Line 92. Changed to ABA phytohormone.
Point 7:line 119: Fertility
Response 7:Line 102. We have carefully checked the manuscript and corrected the errors accordingly.
The Introduction finds itself elaborative. However, it is not systematically structured for the context of the research work done in this paper. It should address
- Significance of Aluminimum toxicity in terms of crop growth/ yield, symptoms in plant etc.
Response 1:Added at the end of the first paragraph of the introduction. Line 48-53. “When plants accumulate excess A13+, the most obvious symptom is inhibited root system [7]. Under Al stress, the uptake of essential elements such as magnesium (Mg), phosphorus (P), iron (Fe) and molybdenum (Mo) through roots was inhibited, resulting in stunted plant development, reduced leaf area, and yellowing and ultimate death of plants [8]. Moreover, the P utilization efficiency and Fe transport efficiency with Al tolerance may be adapted to acidic soils, implying that the molecular mechanisms behind the coevolution are complex [9,10].”
- Complicacy of aluminimum forms in terms of acidic/basic, high / low concentration, interaction with phosphorus, iron etc.
Response 2:Information on the complexity of aluminium in terms of acid/base, high/low concentrations is in the second paragraph of the introduction. Line 54-66.
“Previously, low concentrations of Al3+ were shown to have beneficial effects on a number of plant species, especially those native to the tropics in acidic soils [11]. Low-concentration Al3+ alleviates H+ and P toxicity under low pH conditions and excess P, respectively [12]. The Al3+-promoted growth has also been observed in low-P conditions and in plants tolerant to low pH which means that the phenomenon that Al3+ promotes plant growth also exists in the absence of excess H+ and P [13]. Low concentrations of Al3+ in rice promoted growth and increased soluble sugar content in roots [11]. Soluble sugars might act as osmo-protectants, and low concentrations of Al3+ induced the expression of a sugar transporter (ERD6), which facilitated the transport of soluble sugars to specific tissues or organelles [11]. Thus, low concentrations of Al3+ act as bio-stimulant in crops to promote growth and development. However, there no evidence that whether the peanut generally grown in South China have also evolved such mechanism to promote growth under low-concentration Al3+.”
The complexity of the interaction of Al with phosphorus, iron, etc. is at the end of the first paragraph of the introduction. Line 51-53
“Moreover, the P utilization efficiency and Fe transport efficiency with Al tolerance may be adapted to acidic soils, implying that the molecular mechanisms behind the coevolution are complex [9,10].”
- Signaling of aluminium starting from uptake till growth response, genes known, explaining networks with hormone signaling, any genomic work done
Response 3:In the third paragraph of the introduction. Line 67-98
- Gaps to be addressed in this paper and its significance
Response 4:In the fourth paragraph of the introduction. Line 99-110
Eliminate irrelevant and not so related topics like, other crop tea, peony, alfalfa, batteries, buck wheat etc.
Response: Irrelevant and not-so-relevant topics have been removed, such as other crops tea, peonies, alfalfa, batteries, buckwheat, etc.
Eliminate unnecessary contexts like definition of PCD. Make clear and short sentences, short context on: in which form aluminium is taken up by plant.
Response: Unnecessary context, such as the definition of PCD, has been eliminated. The form in which plants absorb aluminum is in line 46-48.
“When the soil pH is strongly acidic, the EXAl mainly exists in the form of A13+; the most easily adsorbed by negative charges, thereby entering plant cells [6]”
M&M
Point 1: Experiment time frame was only 25 days. However, how many biological replicates are done to generate the samples?
Response 1: In the treatment with different concentrations of Al, because the high concentration Al solution seriously inhibited the growth of peanut seedlings, the seedlings were treated for a total of 25 days. All samples including phenotypic photographs, transcriptomic assays, quantitative gene expression, and hormone assays were collected at two weeks and we have made corrections in line 125. Experiments such as transcriptomic analysis, quantitative gene expression and hormone determination were completed within one year of sampling. Biological replicates under each Al treatment were 3 times.
Point 2: Why morphology (Figure 1) was done in 2 weeks, whereas transcriptome, gene expression was done in 25 days?
Response 2: 25 days are the total time for different concentrations of Al to treat peanut seedlings. All samples including phenotypic photographs, transcriptomic assays, quantitative gene expression, and hormone assays were collected at two weeks and we have made corrections in Line 125.
Point 3: Reviewers link for the NCBI data submission is missing.
Response 3: Raw sequencing data have been uploaded in the NCBI Gene Expression Omnibus under the accession number PRJNA754251 (https://www.ncbi.nlm.nih.gov/sra/?term=PRJNA754251).
Point 4: Experimental details of probes found, DEGs obtained, normalization, filtering criteria etc. missing
Response 4: We've added to lines 131 and 140-150.
Point 5: Calculation of Fold change procedure for the QRT PCR missing
Response 5: We've added to lines 166. “The data were analyzed using the 2−∆∆CT method”.
Point 6: Cartridge details with model no, Detection of ABA, retention time, standards used etc are missing for ABA quantification
Response 6: We've added to lines 191-194. “The CAS number of the ABA standard product was 14375-45-2, and the gradient concentration of the standard sample was 0.2, 0.5, 1, 10, 40, 80, 120, 160, 200, 300 ng ml-1. The retention time of ABA was 3.58 min. The equation was Y=4.441e5X. The fit was 0.9992 [15].”
Results:
Point 1: Line 216: not significantly different: replace with : insignificant
Response 1: Line 212: the expression of the FRDL1 gene was insignificantly different under the Al1 treatment.
Point 2: Figure 1 Legends: A, B: 15 day old plants, C,D: 25 day old plants. Not written clearly in legends. Is this absolute transcription level or relative transcription level? Or Log 2 Fold change?
Response 2: Line 215-221. Figure 1. Effects of different concentrations of Al3+ on the morphological characteristics of peanut and the expression of ALMT9 and FRDL1 genes. Plant height was measured for 14 days of Al treatment (A). Effect of different concentrations of Al on peanut plant height (B). qRT-PCR analysis of relative transcription levels of ALMT9 (C) and FRDL1 (D). Al0, Al1, Al2 and Al3 represent 0 (pH 6.85), 1.25 (pH 4.03), 2.5 (pH 3.85) and 5 (pH 3.69) mmol l-1 AlCl3.18H2O solutions, respectively. Data are means (±SE), n=3. Marking the same letters means P≥0.05 (LSD), there is no significant difference, the difference between different letters means P<0.05 (LSD), and the difference is significant.
Point 3: Line226: Q20 value is not mentioned in Table 1!!!, but mentioned to be present in Table 1. Significance of the Q20 and Q30 values need to be mentioned
Response 3: The Q20 value has been removed from the text. The Q20 and Q30 values represent the number and percentage of bases with sequencing accuracy of 99% and 99.9%, respectively.
Point 4: A statistics of similarity of the replicates from same treatment and dis-similarity of replicates of different treatments is highly required before genomic analysis, it could be shown as higher correlation among the replicates of same treatment and less correlation among data of other treatments.
Response 4: Figure 2. Line 270. Statistical chart of DEGs under different concentrations of Al3+. Red indicates up-regulation, and blue indicates down-regulation (A). Principal component analysis (PCA). The same color represents the same treatment, different colors represent different treatments (B). Sample-to-sample clustering analysis. The red color depicts the upregulation and blue color depicts the downregulation in the gene expression pattern (C). Venn diagram showing overlapping and unique DEGs between different peanut samples Al0-vs.-Al1, Al0-vs.-Al2, and Al0-vs.-Al3 (D).
Point 5:Line 249: Figure S1 is missing but mentioned in text. Figure 2 Legends is completely missing
Response 5: Replace Figure S1 with Figure 2; Replace Figure 2 with Figure 3;
Figure 2. Statistical chart of DEGs under different concentrations of Al3+. Red indicates up-regulation, and blue indicates down-regulation (A). Principal component analysis (PCA). The same color represents the same treatment, different colors represent different treatments (B). Sample-to-sample clustering analysis. The red color depicts the upregulation and blue color depicts the downregulation in the gene expression pattern (C). Venn diagram showing overlapping and unique DEGs between different peanut samples Al0-vs.-Al1, Al0-vs.-Al2, and Al0-vs.-Al3 (D).
Line :281-284. Figure 3 Regulations of the KEGG pathway. Al0, Al1, Al2 and Al3 represent 0 (pH 6.85), 1.25 (pH 4.03), 2.5 (pH 3.85) and 5 (pH 3.69) mmol l-1 AlCl3.18H2O solutions, respectively. Different colors represent different biological processes, and numbers represent the number of DEGs in the same biological process under different Al treatment conditions.
Point 6:Figure 2: Y-axis labels Number of gens missing. A venn-diagram showing DEGs among different treatment sets would be clearer.
Response 6: The number of genes with missing Y-axis labels has been added. Venn diagram is placed in Figure 2D
Point 7:Colour codes are creating confusion due to being similar colour, each histogram bar could be numbered instead. The pathway classification is arranged according to Al 0 Vs Al 1. However, pathway classification of the rest two treatments should also be taken individually not to skip any significant observation in them.
Response 7: Each histogram bar has been numbered. See Figure 3.
Point 8:Line 267, line 270: ….increased by 186.91 % etc. It is better to calculate in how many folds in this case
Section 3.6: similarly anything above 100% could be better explained by fold level.
Response 8: We have been revised.
Point 9: A.hypogea AhNCED1 is involved ABA signaling. However, which NCED here is not mentioned.
Response 9: NCED1
Point 10: All figure legends are not explanatory. It should be elaborated with little sample and experiment description to understand the figure and correlate with the result text.
Response 10: All figure legends are described in detail.
Point 11: Line No 296: consistency of the FC in the genomic data and QPCR data must be shown in table and not mentioned in one line text. In Figure 5B, in addition to the random selection of genes, the Al signalling genes like, ASR, STOP, RAE etc QRTPCR could strengthen the claim the results of this paper
Response 11: The concordance of FC in genomic data and qPCR data is shown in Table 4. The selection of genes in Figure 5B was random, so the expression patterns of ASR, STOP, RAE under different Al treatments are shown in Figure 6A.
Table 4. FC values calculated from randomly selected gene expression and FPKM values obtained by sequencing.
|
Gene |
Sample |
FC |
Gene |
Sample |
FC |
|
OFP1 |
Al0_RNA-seq |
OFP1 |
Al0_qRT-PCR |
||
|
OFP1 |
Al1_RNA-seq |
4.2249179 |
OFP1 |
Al1_qRT-PCR |
4.4242215 |
|
bZIP60 |
Al0_RNA-seq |
bZIP60 |
Al0_qRT-PCR |
||
|
bZIP60 |
Al1_RNA-seq |
1.9782843 |
bZIP60 |
Al1_qRT-PCR |
6.1841118 |
|
bHlH122 |
Al0_RNA-seq |
bHlH122 |
Al0_qRT-PCR |
||
|
bHlH122 |
Al1_RNA-seq |
0.7069601 |
bHlH122 |
Al1_qRT-PCR |
2.0390607 |
|
bHlH18 |
Al0_RNA-seq |
bHlH18 |
Al0_qRT-PCR |
||
|
bHlH18 |
Al1_RNA-seq |
1.4979671 |
bHlH18 |
Al1_qRT-PCR |
3.4426249 |
|
MYB44 |
Al0_RNA-seq |
MYB44 |
Al0_qRT-PCR |
||
|
MYB44 |
Al1_RNA-seq |
2.9794002 |
MYB44 |
Al1_qRT-PCR |
2.4741158 |
|
AP2-AIl5 |
Al0_RNA-seq |
AP2-AIl5 |
Al0_qRT-PCR |
||
|
AP2-AIl5 |
Al1_RNA-seq |
1.1578537 |
AP2-AIl5 |
Al1_qRT-PCR |
2.1525224 |
|
AP2-ANT |
Al0_RNA-seq |
AP2-ANT |
Al0_qRT-PCR |
||
|
AP2-ANT |
Al1_RNA-seq |
0.8790023 |
AP2-ANT |
Al1_qRT-PCR |
0.3706083 |
|
bHlH30 |
Al0_RNA-seq |
bHlH30 |
Al0_qRT-PCR |
||
|
bHlH30 |
Al1_RNA-seq |
1.1577513 |
bHlH30 |
Al1_qRT-PCR |
1.1665719 |
|
NAC56 |
Al0_RNA-seq |
NAC56 |
Al0_qRT-PCR |
||
|
NAC56 |
Al1_RNA-seq |
6.5449812 |
NAC56 |
Al1_qRT-PCR |
20.860103 |
|
NAC29 |
Al0_RNA-seq |
NAC29 |
Al0_qRT-PCR |
||
|
NAC29 |
Al1_RNA-seq |
1.0325376 |
NAC29 |
Al1_qRT-PCR |
15.497673 |
Discussion:
Point 1: The 1st paragraph explains about mechanisms of Al stress tolerance at molecular level, it can be preferable shifted to Intro. Even the 2nd paragraph, line no 329-332: significance of transcriptome bears its relevant place in intro
Response 1: This part is included in the introduction, therefore removed from the discussion.
Point 2: No need to mention figure numbers again in discussion.
Response 2: Figure numbers have been removed from the discussion.
Point 3: line 303: is root production organic acids….: correct grammar as
production of organic acids by roots
Response 3: This section has been deleted.
Point 4: line 310, Line 368: Arabidopsis : capital and italics
Response 4: This section has been deleted.
Point 5:….which have been confirmed…. Replace with correct grammar sentence
Response 5: This section has been deleted.
Point 6: line 316-328: There is a great problem in explanation of the data.
STOP1 expression is shown in Figure 5A. This shows STOP1 is negative FC in Al1 and gradually becomes positive FC in Al2 and Al3. So -ve STOP1 could not induce the ALMT9 in Al1. And why STOP1 and RAE1 QRTPCR is not shown in Figure 5B? Again FDL1 regulation at protein level is not shown in Figure 1D, why there is a mention of protein level here?
Response 6: The selection of genes in Figure 6B was random, so the expression patterns of ASR, STOP, RAE under different Al treatments are shown in Figure 6A. The protein modification levels were mentioned because we thought that FRDL1 might also be transcriptionally regulated by protein modifications like ALMT9, but I removed the protein modification levels to prevent misleading.
Point 7: Line No 333-335: Grammar incorrect, rephrase the sentence. The authors want to say that Al2, 3 are having induced with ABA pathway which is not found in Al1. In that case the pathway analysis for all 8 hormones should be shown in A1,2, 3. In Figure 5A, fold expression FPKM value of all TFs bears no relevance here, rather all the hormone signalling and the Al signalling genes’ heat map would have more claim on data.
Response 7: Line 351-354: “This study showed that the DEGs in the Al0-vs.-Al1 sample group were significantly enriched in starch and sucrose metabolic pathways. The DEGs in Al0-vs.-Al2 and Al0-vs.-Al3 sample group were significantly enriched in plant hormone signal transduction pathway.”
We measured the other four hormones and did not show the same pattern as ABA (Fig. S1). In Figure 6, we express the relationship between transcription factors and peanut aluminum tolerance, so it is a heat map of transcription factors. The expression levels of Al signaling core genes (ALMT9 and FRDL1) have been determined (Fig. 1), and the contents of other four hormones have been determined. We believe that under different Al treatments, the determination of hormone levels is more convincing.
Figure S1. Effect of different Al3+ concentrations on the plant hormone content. The effect of Al treatments on salicylic content (A). The effect of Al treatments on gibberellin content (B). The effect of Al treatments on jamonic acid content (C). The effect of Al treatments on indoleacetic acid content (D). Al0, Al1, Al2 and Al3 represent 0 (pH 6.85), 1.25 (pH 4.03), 2.5 (pH 3.85) and 5 (pH 3.69) mmol l-1 AlCl3.18H2O solutions, respectively. Data are means (±SE), n=3. Marking the same letters means P≥0.05 (LSD), there is no significant difference, the difference between different letters means P<0.05 (LSD), and the difference is significant.
Point 8: Wrong mention of Reference 46: line No: 348
Response 8: I apologize for our carelessness, all references have been carefully checked and confirmed to be accurate.
Point 9: ABA induced the expression of AtALMT1….. is shown in Kobayashi et al. 2013, Reference Number 43.
Response 9: I apologize for our carelessness, all references have been carefully checked and confirmed to be accurate.
Point 10: Line 348-352: the data and the statements are not enough to claim that “ABA improved Al tolerance by modulating organic acids.” Again Reference 47 doesn’t explain “ABRE existed in the VUMATE promoter region”. It is a wrong mention of reference. Together there is not enough data to claim higher ABA content and gene expression regulated organic acid secretion. For this claim, more data required to show in terms of organic acid secretion.
Response 10: Line 365-371. The current study can only show that ABA is able to link Al signaling with phytohormone signaling, it cannot yet show that ABA improves aluminum tolerance by regulating organic acids, we have made revisions.
Point 11: Line 355-358: great confusion in explaining data.
If Al2,3 show Aluminium tolerance by increasing ABA content, then why it is not visible in morphology of the Al2,3? So increase in ABA content and NCED expression is for Al tolerance phenotype or for sensitive phenotype?
Response 11: This study found that high concentrations of Al2 and Al3 promoted the up-regulation of ABA content in peanut leaves, while low concentrations of Al1 did not increase ABA content, indicating that high concentrations of Al2 and Al3 made peanuts sensitive to Al, induced the expression of NCED1, and increased endogenous ABA content.
Point 12: Line 364: again there is wrong mention of Referenc 51, I think the ASR1 and ASR5 reference is present in 47.
Response 12: I apologize for our carelessness, all references have been carefully checked and confirmed to be accurate.
Point 13: I think the authors should carefully correlate the whole reference list with the text and renumber them again.
Response 13: I apologize for our carelessness, all references have been carefully checked and confirmed to be accurate.
Point 14: Line 359-376: This paragraph discusses basically Figure 5A, heatmap, fold expression of TFs. It should be actually discussed in the result section first. Moreover, In the discussion part, along with other’s references, the present data should be discussed.
Response 14: Line 376-398
“In present study, it was found that the expression of a large number of TFs was different in different concentrations of Al3+ treatment in peanut. Genes encoding bHLH, NAC and STOP1 have negative FC in Al1 and gradually become positive FC in Al2 and Al3. The WRKY46 could negatively regulate the expression of ALMT1 under Al stress, but the WRKY22 could positively regulate the expression of FRDL4 under Al stress in rice [42]. The expression of ASR was induced by ABA, and the expression of ASR1 and ASR5 genes was also induced by Al stress [43]. The ASR1 and ASR5 have complementary roles, specifically binding to a cis-acting element in the STAR1 (sensitive to Al rhizotoxicity) promoter to activate STAR1 expression [44]. The STAR1 and STAR2 encoded ATP-binding proteins that transported UDP-glucose and masked Al-binding sites in the cell wall [44]. The GsMAS1 was a MADS-box transcription factor, and overexpression of GsMAS1 in Arabidopsis could improve Al tolerance [45]. The Al treatment induced the expression of the NAC gene in rice, and Al promoted its growth by mediating the expression of NAC [46]. The ABI5 positively regulated the tolerance against Al stress in rice [41]. The STOP1 and ART1 could positively regulate the expression of genes responsive to Al stress (FRDL and ALMT, etc.), but the genes encoding STOP1 and ART1 did not respond to Al-induced expression [47]. The zinc finger transcription factor STOP1 was a key transcription factor regulating the expression of AtALMT1, which may be regulated by Al at the post-transcriptional or post-translational level [23,32,33]. The ART1 and STOP1 play a very important role in regulating Al tolerance in plant. Studies have found that after the ART1 and STOP1 fragments in tobacco (Nicotiana tabacum) and moss (Physocmitrella patens) were knocked out, plants were observed to be sensitive to Al3+ [18]. Therefore, The TFs played an important role in Al regulation of peanut growth and development.”
